# The Evaluation of Village Fund Policy in Penukal Abab Lematang Ilir Regency (PALI), South Sumatera, Indonesia

**Suandi** [1,2,*], **Entang Adhy Muhtar** [1], **Rd Ahmad Buchari** [1] **and Darto** [1]

[1] Department of Public Administration, Faculty of Social and Political Science, Universitas Padjadjaran, Jl. Raya Bandung-Sumedang Km. 21, Jatinangor, Sumedang 45363, Indonesia
[2] Department of Public Administration, Faculty of Administrative Science, Universitas Sjakhyakirti, Jl. Sultan M. Mansyur, 32 Ilir, Kebon Gede, Palembang 30145, Indonesia
* Correspondence: suandi19001@mail.unpad.ac.id

**Abstract:** This study reported the village fund evaluation of Penukal Abab Lematang Ilir regency (PALI), South Sumatera, Indonesia. The amount of village funds in the Penukal Abab Lematang Ilir regency (PALI) has increased yearly. However, it was also acknowledged that this has not been enough to impact PALI regency significantly. This study used qualitative mixed methods, and the data were collected using interviews, documentation and observations with nine informants from four selected villages, including Babat village, with a developed village typology, Muara Sungai village and Tanah Abang Selatan village, with an underdeveloped village typology, and Karta Dewa village, with a developing village typology. The research showed the village funds were divided into four indicators in the local government's commitment to developing Indonesia from the periphery by strengthening regions and villages within the framework of a unitary state. However, the village fund policy evaluation in PALI regency, South Sumatera Province, has not been practical in regards to input, process, output, and outcome indicators. In addition, this paper provided insight into the development and village innovation field to evaluate village funds.

**Keywords:** evaluation; village fund; policy; PALI regency; South Sumatera



## 1. Introduction

Village development has a crucial and strategic role in the frameworks of national development and regional development because it contains elements of equitable distribution of development and its results, and directly interacts with the interests of the majority of people who live in rural areas to improve their welfare. The village is the lowest governmental structure in the Indonesian government system and is also a government entity directly related to the people. The future of a country lies and depends on the success of a village [1,2]. The Indonesian government emphasizes that development starts from the periphery by strengthening rural areas in Indonesian territory. This development started with data collection on village potentials, carried out three times in ten years. The data collection results showed an increase in 2018 of 83,931 from 82,190 (2014) and 78,609 (2011). In addition to data collection on village potential, data collection on the population of cities and villages is also compared. It showed that the number of residents in rural areas was greater than in urban areas. For this reason, it was necessary to develop the village. Sustainable village development will make a village develop in all aspects, especially aspects of infrastructure, which are the needs of rural communities [3,4].

Law Number 6 of 2014 was published discussing the village, further supporting Government Regulation Number 43 of 2014 regarding implementing regulations (Undang—Undang Nomor 6 Tahun 2014). This law provided opportunities for rural communities to regulate and manage their households, taking into account the principles of democracy, community participation, equity, and justice, as well as paying attention to regional potential and diversity [5].

One of the Indonesian government's policies in supporting village development was by issuing village funds as a source of village income. This policy has been implemented worldwide, including in Thailand, which provided microcredit to the poorer, agricultural households, but there was abuse in its implementation [6]. Besides Thailand, Myanmar also has a policy of local development funds (LDFs), which focuses more on government planning and budgeting. Specifically, LDFs are used to identify and fund small-scale infrastructure projects such as bridges and roads connecting villages, drainage, drinking water and irrigation projects [7]. In Laos, village development funds, or VDCs, were soft-assistance-sourced from the World Bank and the Finnish Government [8].

This policy was undoubtedly different from the village fund policy implemented in Indonesia. According to Government Regulation Number 8 of 2016, the Second Amendment to Government Regulation Number 60 of 2014 concerning village funds sourced from the Article 1 State Revenue and Expenditure Budget was intended to finance infrastructure development and included community empowerment. The village fund was expected to provide additional energy for villages to be more robust, advanced, and independent and reduce hunger [9–11]. The policies implemented since the 2015 fiscal year have resulted in significant development in Indonesia. The village fund policy's successful implementation by the government is increasingly gaining international recognition. This can be seen from the appreciation given by the World Bank [12,13]. A total of 23 developing and poor countries have implemented the village fund model. In line with the success of village funds, the number of village funds in Indonesia continued to reduce the number of poor rural people from 17.8 million (14.2%) in 2015 to 15.8 million (13.2%) in 2019. In addition, village funds have also succeeded in improving the status of villages. In 2014, there were 19,750 underdeveloped villages, while 13,232 underdeveloped towns, developing villages and independent villages in 2018 (Indeks Pembangunan Desa 2019). The Corruption Eradication Commission explained that the village fund has four weaknesses: namely, regulation, management, supervision, and the human resources that manage village funds (https://news.detik.com/, accessed on 2 September 2022). The main problem is coaching and leadership [14].

South Sumatra is one of the provinces whose villages receive village funds. The amount of village funds for villages in the South Sumatra province is increasing yearly. The village funds reached 13 districts and one city in South Sumatra, distributed between 232 sub-districts and 2859 villages. However, in 2018 this was reduced to 2853 villages because six villages in Empat Lawang regency had become kelurahan. Penukal Abab Lematang Ilir regency (PALI) is one of the regencies in South Sumatra Province. Penukal Abab Lematang Ilir regency (PALI) is a new autonomous region (DOB) resulting from the division of Muara Enim regency, which was ratified on 11 January 2013 through Law No. 7 of 2013.

The amount of village funds in the Penukal Abab Lematang Ilir regency has increased yearly. The PALI regency received village fund budgets starting in 2016. The amount of village funds received by the PALI regency in 2016 was IDR 47,604,942 compared to IDR 60,334,554 in 2017, IDR 63,961,929 in 2018, and IDR 77,195,208 in 2019. These data show that village funds continue to increase yearly (http://www.djpk.depkeu.go.id/, accessed on 29 August 2022). Based on the preliminary research that the authors have carried out, although the amount of funds in the PALI regency continues to increase, it is also recognized that this has not been enough to significantly impact the regency.

It can be seen from the HDI (Human Development Index) of the PALI regency that the IPM of Penukal Abab Lematang Ilir regency in 2019 was 64.33, ranking 16th out of the 17 regencies/cities in South Sumatra Province. HDI is an index that explains how the population can access development outcomes in obtaining income, health, education, etc. This means that the HDI of the PALI regency is still relatively low. In addition, based on the IDM (Independent Village Index), from 2014 to 2018 the PALI regency did not yet have an independent village. For Maju village alone, there was no increase in the number of developed towns. Not only that, but the Village Development Index (IPD) has only

reached the developing stage, and there are no independent villages. The data can be seen in Figure 1.

**Figure 1.** Independent Village Index (IDM) and Village Development Index (IPD) in 2014 and 2018 (Source: bps.palikab.go.id 2021 [15]).

Previous research has explained that the first indication showed that the many constantly changing regulations confused village officials and the community. The second indication was the lack of community involvement in the planning and implementation village fund policies. This results in a reduced sense of ownership. Based on the movements mentioned, this meant that the evaluation of the village fund policy carried out from 2015 to 2019 did not solve all the problems of the village fund. It was necessary to evaluate the village fund policy to improve rural communities' welfare and to reduce poverty with global development, referred to as the transformation of Sustainable Development Goals (SDGs) [16]. Due to the urgency, this research was more about reviewing and evaluating village fund policies, including input, process, output, and outcomes, and obtaining a new concept regarding the evaluation of the village fund policy in the Penukal Abab Lematang Ilir (PALI) regency, South Sumatra Province.

## 2. Literature Review

### 2.1. Public Policy

Public policy is defined as the authoritative allocation of values for a society as a whole, or as the coercive allocation of values to all members of society. Public policy is a hypothesis that contains initial conditions and predictable consequences and can be distinguished from other policies [17–19]. The involvement of non-government factors influences this. There was also a definition of public policy as "the relationship between government units and their environment" [20], but also a set of general tools and mechanisms that provided choices of action to be made and whatever actions the government "chooses to do or not to do" [21–23].

The public policy contained several decisions that were chained (not single decisions, but many non-separate choices) [24] and the objectives were clear, including ways to achieve these goals, made to respond to problems that occurred in a particular situation by an actor or many political actors (executive, legislative and judiciary, including non-governmental actors). The policies that have been set must be reviewed through public policy evaluations [1,25,26].

*2.2. Policy Evaluation*

Evaluation is considered a governance tool used by policymakers and a learning tool to generate future practice [27–29]. Policy evaluation is an objective, systematic and empirical examination of the effects of public policies and programs on their targets in terms of the objectives to be achieved [21]. Policy evaluation is an activity that involves the estimation or assessment of policies that include substance, implementation and impact [1].

Based on opinion, evaluation characteristics were divided into four categories [7]. First, a value focus determines the benefits or social utility of the policy. The evaluation included procedures for evaluating the goals and objectives themselves. Second, the interdependence of value facts that determine a policy's success must have evidence. Claiming that a particular policy or program has achieved the highest (or lowest) level of performance requires not only that the policy outcomes are valuable to all individuals, groups or entire societies, but that it must be supported by evidence that policy outcomes are a consequence of actions taken to solve particular problems. Third, in the present and past orientations, evaluative demands differ from advocating demands directed at present and past results rather than future outcomes. Fourth, the duality of values, namely evaluations with multiple qualities, are seen as both an end and a means [30,31]. The above policy evaluation informs us that policy evaluation is not just collecting information about policies that can be anticipated and which cannot be anticipated, but the evaluation is directed to provide information on the past, present and future. In addition, policy evaluation is required to state that a particular policy or program has achieved the highest (or lowest) performance for all individuals, groups and communities if actions are taken to address the problem.

In this study, the evaluation measurements used the criteria of input, process, output and outcomes [32]. Input indicators are resources used to implement policies needed so that the implementation of activities can produce the specified output, e.g., funds, human resources, and information [33]. Process indicators were all quantities carried out to process inputs into outputs through direct services to the community. Output indicators were expected to be directly achieved from an activity that could be physical or non-physical. Outcome indicators were actions taken by the target group and the consequences of those actions [34].

## 3. Description of the Case Study Area

This study will evaluate the village fund policy in Penukal Abab Lematang Ilir (PALI) regency, South Sumatra province. Penukal Abab Lematang Ilir regency (PALI) has an area of 1840.0 square kilometers (km$^2$) and is ranked 13th out of 17 regencies/cities in South Sumatra province. The capital city of the Penukal Abab Lematang Ilir (PALI) district is Talang Ubi. The geographical position of Penukal Abab Lematang Ilir (PALI) regency is located in a lowland area with an altitude between 16–84 m above sea level (see Figure 2). Based on the altitude distribution area by sub-district, the entire area is located at an altitude of fewer than 100 m above sea level. This area is directly adjacent to Musi Banyuasin regency and Banyuasin regency to the north; Muara Enim regency and Prabumulih City in the south; Muara Enim regency in the east; and Musi Rawas regency in the west. The area of Penukal Abab Lematang Ilir (PALI) is classified as agrarian, with average rainfall in 2019 varying between 25 mm to 320 mm, where the highest rainfall occurs in February. There are four types of soil in Penukal Abab Lematang Ilir (PALI): alluvial, red–yellow podzolic, gley association, and yellowish–brown podzolic association. These four types of soil are found in almost all sub-districts in Penukal Abab Lematang Ilir (PALI) regency, except for yellowish brown podlosic association soil, which is only found in Penukal District.

Based on the results of the annual population census, the population of the Penukal Abab Lematang Ilir (PALI) regency was recorded to be 187,281 people in 2018, 189,764 in 2019, and 192,199 in 2020 [15]. From this data, it can be seen that in 2019 the population growth rate in the Penukal Abab Lematang Ilir (PALI) regency was recorded at 1.32 percent compared to the previous year. This indicates an increase in the population in this region,

from 187,281 people in 2018 to 189,764 people in 2019. In addition, population growth in 2019 experienced a slowdown compared to the previous year, with a growth rate of 1.32 percent. Most of the population (41.92%) live in the district capital, Talang Ubi.

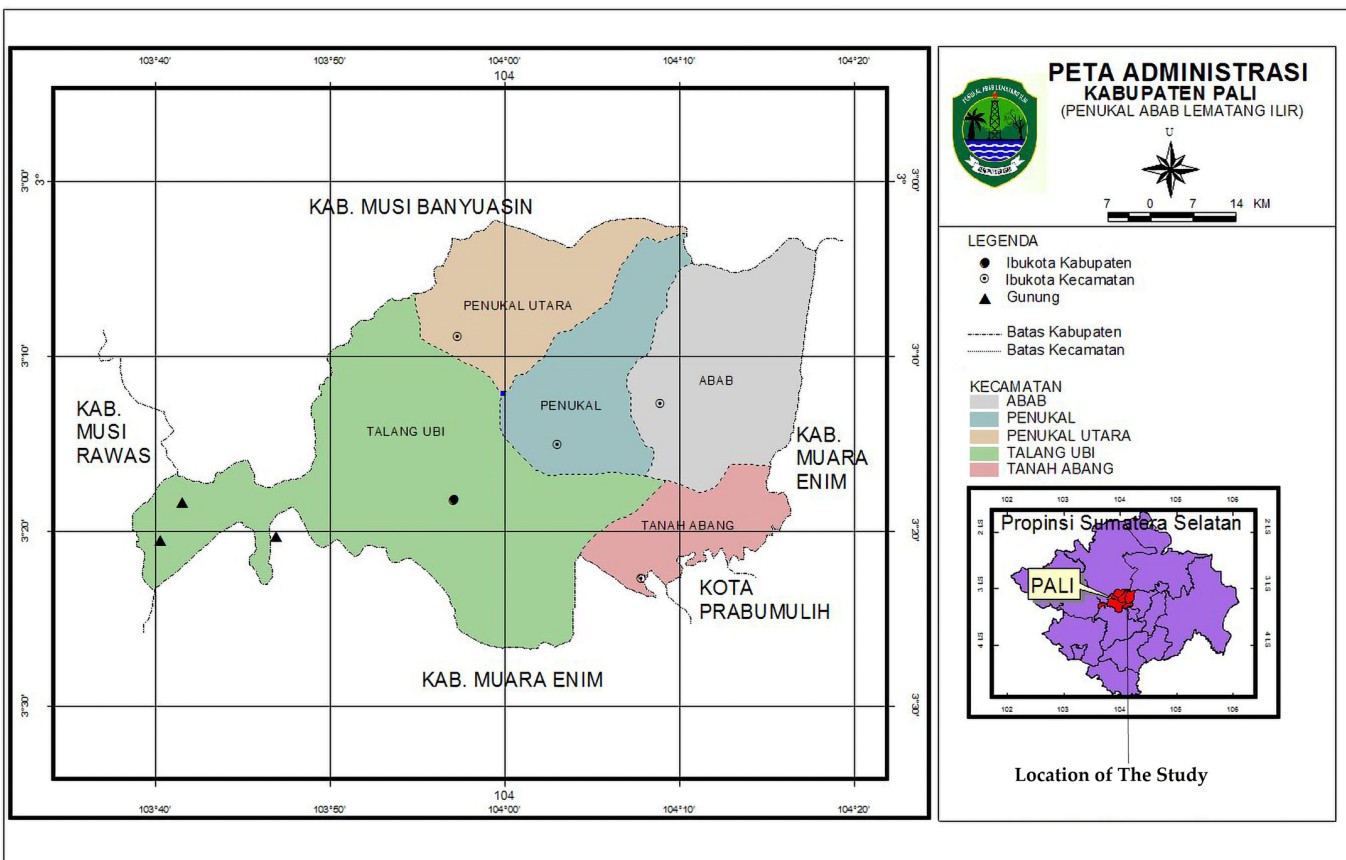

**Figure 2.** Geographical setting of Penukal Abab Lematang Ilir (sources: Maplands, edited by researcher).

Furthermore, when viewed in terms of age group, the population of children (0–14 Years) in the Penukal Abab Lematang Ilir (PALI) regency in 2018 reached 303.05%, while the productive age population reached 62.8%. The age were 4.12% of the total population in the regency of Penukal Abab Lematang Ilir (PALI). Likewise, in 2019, the child population had a percentage of 32.8%, productive age reached 62.9% and older adults were 4.2%. It can be seen that the population of the Penukal Abab Lematang Ilir (PALI) regency is generally productive. The composition of a population like this is beneficial if the population is qualified and can contribute to development. Vice versa, this can also be detrimental if the population is of low quality. Therefore, the government of Penukal Abab Lematang Ilir (PALI) must provide maximum education because only then can the population be used to the maximum. Besides that, it can also be stated that the overall male population in Penukal Abab Lematang Ilir (PALI) regency is higher than the female population.

The economic structure in Penukal Abab Lematang Ilir (PALI) in 2019 was still dominated by the primary sector, particularly the mining and quarrying category (43.20 percent) and the agriculture, forestry and fisheries category (15.24 percent). At the same time, the second largest role was the tertiary category, which comes from the wholesale and retail trade category; the repair of cars and motorcycles (14.65), while in the tertiary category, the most considerable role was the construction category (14.34) [15].

The PMK Number 49/PMK.07/2016, an evaluation of the village fund policy, was carried out using the calculation of the distribution of the amount of the village fund for each village by the regency/city, and the realization of the distribution and use of the village fund, assessed based on appropriate inputs, processes, outputs, and outcomes, and evaluated based on proper inputs, processes, outputs, and outcomes [32].

## 4. Design of the Research

This study used mixed methods, such as qualitative research focusing on interviews and observations, to gather data from nine informants (N = 9) [35]. These informants are needed to determine the conditions following the evaluation of the village fund policy. Therefore, the informants in this study were the formulators, policy implementers and policy target groups who understand and are involved in the village fund policy in Penukal Abab Lematang Ilir (PALI) regency, with key informants (Key Informants) being part of the village apparatus in three districts of the Penukal regency Abab Lematang Ilir. The informants were the village Community Empowerment Agency (BPMD) employees and the subdistrict head. Data were collected through in-depth interviews. Before field-work, some technical interview exercises and field observation were carried out to give related information on the details of the number of village funds, distribution and use of village funds.

Another method used was the descriptive method to obtain data in the field when conducting research [36]. The unit of analysis in this study was the Government of Penukal Abab Lematang Ilir regency (PALI), which consists of 5 (five) and 65 (sixty-five) villages. As for the villages taken for each sub-district, villages with IPD status were left behind and developed, then 4 (four) villages were selected: Tanah Abang Selatan village, with disadvantaged village status; Muara Sungai village, with underdeveloped village status; and Karta village and Tripe village, with developed village status. Data collection techniques require evaluation or research information from previous research and documents. The documents required for this study were related to determining the details of the number of village funds and the distribution and use of village funds in the Penukal Abab Lematang Ilir (PALI) regency. Data validity techniques must be carried out to ensure that research efforts can be justified. In this study, the validity technique used was triangulation. Triangulation is a technique for checking the validity of data that utilize Secondary data for matching or as a comparison against that data [37]. The data analysis technique used in this study was a qualitative descriptive analysis technique using an analytical model [38].

## 5. Results and Discussion

One of the policies issued by the government of Indonesia in the context of equitable distribution and acceleration of national development is the village fund policy [2,5]. Through this policy, the government is building Indonesia from the periphery. In Government Regulation Number 60 of 2014, village funds were sourced from the State Revenue and Expenditure Budget designated for villages, which were transferred through the district/city Regional Revenue and Expenditure Budget and were used to finance government administration, implementation of development, community development, and empowerment of the public. The extreme current attention of the government towards villages was evidenced by the revolving of village funds, which began in 2015 with a budget from the APBN of IDR 10.7 trillion. The village's budget increased yearly [12,39]. The priority of village funds in financing village development was aimed at improving the welfare of the village community, improving the quality of human life and reducing poverty (P.P. No. 60 of 2014).

Penukal Abab Lematang Ilir (PALI) regency is one of the regencies that received village funds for villages under the PALI regency government. Regulations regarding the distribution and amount of village funds for each village in the regency (PALI) were regulated in Perbup No. 1 of 2019 concerning procedures for distribution and determination of fund details for each village for the 2019 fiscal year. In 2018, PALI received village funds of IDR 63.96 billion and allocated them to 65 villages spread over five sub-districts. Then, in 2019, this village fund increased to IDR 77.19 billion following the central government's policy of increasing village funds.

1. District Inspectorate;
2. Community and Village Empowerment Service (DPMD);
3. District.

The village fund policy evaluation was carried out based on a portfolio assessment by desk evaluation and field assessment. The portfolio assessment was carried out based on the data contained in the Village Fund Policy Consolidation Report. Meanwhile, the field assessment of the evaluator team was carried out by directly observing the villages that received the village fund disbursement.

In specific terms of evaluating the village fund policy in the Penukal Abab Lematang Ilir regency, the results show that the Penukal Abab Lematang Ilir regency has not been optimal in implementing the village fund policy. The evaluation results require clarification and more objective evaluation efforts, considering that several weaknesses and problems are still being found, especially when viewed in terms of the aspects of substance, implementation and results of village fund policies in improving community welfare.

In this section, we will describe the results of the field research and its discussion. In this discussion, the theory used as an analytical tool or guidance was the theory of Bridgman and Davis with several indicators, namely input, process, output, and outcomes. Furthermore, these indicators serve as a guide for researchers in obtaining field data and discussing research results. The results of the field research and discussion used the theory of Bridgman and Davis with several indicators, namely input, process, output, and outcomes [32].

### 5.1. Input Indicators

The inputs in this study were further examined through aspects, regulations, targets, adequacy of funds, facilities and infrastructure, and human resources. The inputs were presented as follows.

### 5.2. Regulations

Regulation is an indicator related to the village funds policies that were conceived based on Law Number 6 of 2014 concerning villages. "The birth of the Law on Villages, namely Law number 6 of 2014". (Head of Penukal District). The use of village funds was regulated in Government Regulation Number 43 of 2014 concerning the amount or percentage of village expenditures in the village budget. According to Permendes Number 19 of 2017, village funds were sourced funds from the State Revenue and Expenditure Budget designated for villages, transferred through the Regency/City Regional Revenue and Expenditure Budget, and were used to fund government administration, development implementation, community development, and community empowerment. "This village fund policy is supported by three ministries, the ministry of finance, the ministry of home affairs, and the ministry of rural areas. Each ministry issues its regulations to regulate village funds" (Head of Tanah Abang District).

Based on information obtained from the Head of Institutional and Community Training, there was often an overlap between the regulations issued by the three ministries that oversee the village fund policy. This can be seen in the Ministry of Home Affairs and the Ministry of Villages each giving their own technical guidelines/rules. This can confuse the village apparatus in implementing this village fund policy. The analysis above showed that the regulations were complete but still needed to be refined to overcome emerging obstacles such as overlapping regulations.

### 5.3. Adequacy of Funds

Every village in Indonesia has the opportunity to receive IDR 1.4 billion annually [3,30]. Penukal Abab Lematang Ilir regency (PALI) received a village fund budget starting in 2016. The amount of village funds received by the PALI regency in 2016 was IDR 47,604,942 compared to IDR 60,334,554 in 2017, IDR 63,961,929 in 2018, and IDR 77,195,208 in 2019. These data show that village funds continued to increase from year to year (http://www.djpk.depkeu.go.id/, accessed on 20 August 2022). The funds provided were insufficient to carry out development activities in the village because several villages have a high population density. This was in line with the following interview with the Karta Dewa

Village Head: "Yes, it is not enough because, in that village, there is much to be done, both physical development and community development".

This meant that the village funds were insufficient for all activities to be carried out. Although many considered the amount of these funds to be inadequate, some believed that this amount was sufficient: "If it is enough, it depends on the skill of the village apparatus in placing the posts. If we think it is more than enough" (Talang Ubi District).

Based on the data analysis above of the adequacy of funds, it can be judged that village funds were sufficient for village activities depending on the expertise of village officials and the community in managing their finances.

### 5.4. Facilities and Infrastructures

This factor is necessary to support the evaluation of the village fund policy in the Penukal Abab Lematang Ilir (PALI) regency. All informants agreed that the government had not provided sufficient equipment to implement the village fund policy for this factor: "There are no facilities from the central government" (Head of Karta Dewa village). "It is not sufficient. Moreover, we are only given funds and are not equipped with facilities. We are only given the siskeudes application. In our opinion, this application is not optimal because its use can be more complicated than manual reports. We hope that if this siskeudes is continued, it can be simplified but detailed" (Head of Babat village).

Based on the analysis above, facilities and infrastructures must be better prepared, because the government did not prepare facilities and infrastructure to implement village fund policies, such as clear information and inadequate I.T. and office equipment availability. In 2019, the government launched the Siskeudes application to assist village officials in reporting village funds.

### 5.5. Human Resources

Human resources refers to an employee or implementor who implements the village fund policy in the PALI regency. The Head of Community Empowerment and Appropriate Technology stated that the existing human resources personnel have inadequate knowledge, thus affecting performance in managing village funds by the principles of good governance.

Based on the description above, the village apparatus' human resource aspect still needed improvement. Improving the quality of human resources can be carried out by providing technical training and guidance and increasing the minimum requirements to become village heads and village officials. Choosing the village head and village apparatus must be based on the skills and knowledge of the candidate for the village head and village apparatus, not only based on the reciprocation factor and those closest to them.

### 5.6. Process Indicator

This indicator covered four categories: planning, implementation, administration and accountability reporting. The planning was carried out according to the policy implementation guidelines, but community participation needs to increase. This also accords with the researchers' observations at the MusrenbangDes activities in Tanah Abang Selatan village. To start MusrenbangDes activities in this village, village officials must find people who want to participate. The meeting was postponed for half an hour to wait for the presence of residents. This means that community participation in voicing their aspirations is still low. The village government should try to find a solution so the community can participate in the reflection. Some villages have tried to invite the community through socialization, while others have used a face-to-face approach with transportation money, as per an interview with the Head of Institutional and Community Training (DPMD) and the Head of the Penukal Sub-district.

It was the same with the implementation that has been going well. The role of the Village Consultative Body has been active in supervising the implementation of the use of village funds. Supporting documents such as village activity plans and budgets, village activity work plans, and budget plans were available in all research sample villages.

However, there were obstacles in determining the priority of use, where not all activities can be carried out using village funds. This was due to the limitations of village funds in meeting village needs. The budget for the use of village funds is formulated from the results of interviews and can be seen in Figure 3 and Table 1.

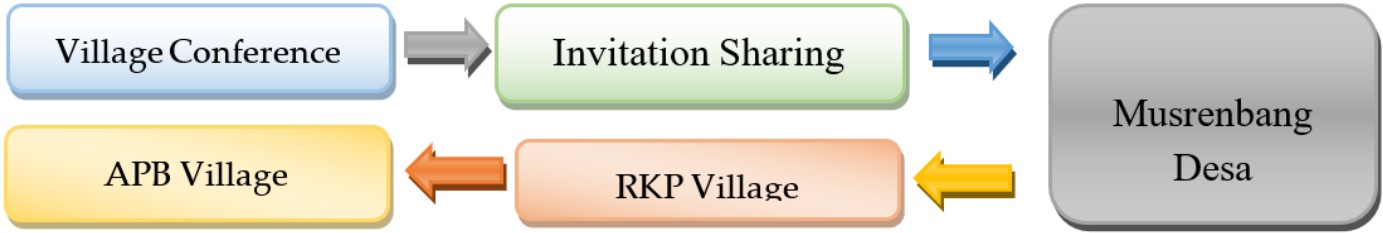

**Figure 3.** Flow of planning for the use of the village fund budget. Source: (Documents, study and interview, 2022).

**Table 1.** Results of the analysis of planning for the use of village funds in Penukal Abab Lematang Ilir district.

| No. | Step | Actor | Analysis |
|---|---|---|---|
| 1 | Village Conference | Head of Hamlet, Community, and Community Leaders | The village government's village fund budget planning process involves the participation of all components in the village, both community institutions and the general public. |
| 2 | Invitation Sharing | Village Apparatus | Village officials distributed invitations to all elements of society, community leaders, NGOs, and BPDs, to attend the Village Musrenbang process. |
| 3 | Musrenbang Village | Camat, Village Head, BPD, NGOs, Community Leaders, and Community. | ✓ The village government received a proposed work program from all elements present in the musrenbang process. ✓ Community participation is not yet significant. |

The administration went well. All sampled villages routinely reported village finances to the village head every month, carried out at the end of each month when the village treasurer closed the books for the current month, and reports were collected on time. The submitted reports were also transparent and accountable and participatory, implemented by involving the entire community. However, there were obstacles in making reports using the Siskeudes application, namely limited skills in using technological tools, so operators were needed. Reporting and accountability were carried out properly, and all accountability report documents were available. However, there were obstacles as several villages did not do this.

*5.7. Output Indicator*

Output is the result of a policy evaluation, whether the implementation of the policy produces outputs or products that are under the stated policy objectives or not. The results or products resulting from this policy were physical development, community development and community empowerment. The village fund received by the Penukal Abab Lematang Ilir district for the 2018 fiscal year was IDR 65,043,635,000, and in 2019 it was IDR 77,233,798,989. Village fund receipts were received gradually rather than all at once. Village fund distribution was carried out in three stages: namely 20% (twenty percent) in stage I, 40% (forty percent) in stage II, and 40% (forty percent) in stage III. Each village must collect the determined requirements to receive village fund distribution (as can be seen in Table 2). The percentage of disbursement was still the same as in the previous year in 2019, namely, phase I of 20% at the earliest in January and no later than the third week of June, stage II of 40% at the earliest in March and no later than the fourth week of June, and stage III by 40% at the earliest in July (Table 3). This is based on the Regulation of the Minister of Finance Number 193 of 2018.

**Table 2.** Data on the ceiling and distribution of village funds in the sample villages in 2018.

| No. | Village Names | Ceiling | Distribution | | |
|-----|---------------|---------|--------------|--------------|--------------|
| | | | Phase 1 | Phase 2 | Phase 3 |
| 1 | Muara Sungai | 758,289,000 | 151,657,800 | 303,315,600 | 303,315,600 |
| 2 | Tanah Abang Selatan | 1,106,524,000 | 221,304,800 | 442,609,600 | 442,609,600 |
| 3 | Karta Dewa | 996,695,000 | 199,339,000 | 398,678,000 | 398,678,000 |
| 4 | Babat | 868,941,000 | 173,788,200 | 347,576,400 | 347,576,400 |

**Table 3.** Data on the ceiling and distribution of village funds in the sample villages in 2019.

| No. | Village Names | Ceiling | Distribution | | |
|-----|---------------|---------|--------------|--------------|--------------|
| | | | Phase 1 | Phase 2 | Phase 3 |
| 1 | Muara Sungai | 881,635,000 | 176,327,000 | 352,654,000 | 352,654,000 |
| 2 | Tanah Abang Selatan | 1,584,995,000 | 316,999,000 | 633,998,000 | 633,998,000 |
| 3 | Karta Dewa | 1,002,102,000 | 200,420,400 | 400,840,800 | 400,840,800 |
| 4 | Babat | 1,042,432,000 | 208,486,400 | 416,972,800 | 416,972,800 |

The timeliness of the disbursement of funds that have been budgeted for the smooth running of a policy, in this case, infrastructure development, cannot be separated from the timely disbursement of village funds. If village funds are disbursed on time and according to the village work activity plan, it will help accelerate infrastructure development to be completed more quickly without having to experience delays. The village development process is a mechanism from the community's wishes that is integrated with the community. The combination that determines the success of the development is a harmonious combination of community participation activities on the one hand and government activities on the other.

The output of village funds was in the form of physical development in Penukal Abab Lematang Ilir regency in several villages, sampled in this study as follows:

- Karta Dewa village focused more on physical development in 2018–2019 on road facilities and infrastructures, such as a total road construction of 888 m, including farm roads and village roads, the rehabilitation of 35 bridge units, the construction of one building unit, two units of stalls and road infrastructure of 500 m. This showed a focused on roads to facilitate community mobility to improve the community's economy.

- Physical development in Tanah Abang Selatan village in 2018 was focused on building PAUD. In addition to the construction of PAUD, waste facilities, stalls, and billboards were also constructed. This PAUD development aimed to improve education in Tanah Abang Selatan village. In addition to the teachers, this PAUD came from the villagers who had the ability and skills in the field of PAUD education. In other words, it also aimed to empower the South Tanah Abang community. In 2019, the village was also still focused on equipping PAUD facilities, but the construction of a 735 m long road and 525 m road infrastructure, as well as the construction of a village cemetery, were also undertaken.

- Development in Muara Sungai village was only focused on road construction. Over two years, this village built a 2200 m long road consisting of a farm road and a village road. Based on observations in several villages, this village is slightly behind compared to other villages; from 2018 to 2019, only road construction was carried out.

- Development in Babat village was also still focused on building a total of 823 m roads and a total of 173 m road infrastructure. In addition to building roads, one unit of bridge rehabilitation was also carried out. This village was slightly different from other villages. This village realized village funds for constructing a village library as much as one unit, and constructing sports infrastructure facilities as much as one unit. This village also realized the manufacture of billboards every year. This meant that this village had good transparency.

In addition to physical outputs such as infrastructure development, the village fund is used in the fields of education, training and counseling for village officials, and the village consultative body, consisting of capacity-building for village apparatus. Based on the technical guidelines for the priority for the use of village funds for the 2018 and 2019 fiscal years, the intended community empowerment was the empowerment of the village community, not the village apparatus.

*5.8. Outcomes Indicator*

Outcomes refer to whether a policy evaluation has a tangible impact on the target group following the policy objectives. To find out the impact resulting from the village fund policy evaluation results in the PALI regency, the researchers looked at the achievement of the village fund policy objectives. The village fund and the increasing amount and increasingly diverse uses in each village, regency/City, of course, can be understood to produce impacts that vary between regions. In general, the village fund policy aimed to improve public services in villages, alleviate poverty, advance the village economy, overcome development gaps between villages, and strengthen village communities as subjects of development.

The outcomes of this village fund policy provided a decrease in the percentage of poverty in the PALI district, as seen in Figure 4, which shows a graph of poverty in the PALI district from 2018 to 2019.

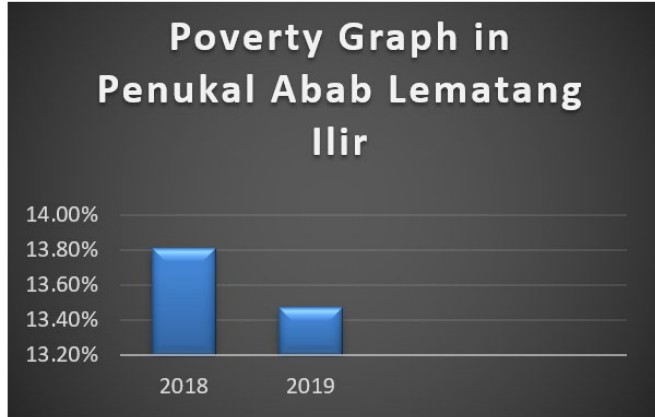

**Figure 4.** Poverty graph of Penukal Abab Lematang Ilir district (BPS Pali 2022) [15].

Based on the graph, in 2018, the poverty rate was 25.78 thousand people with a percentage of 13.81%, while in 2019, the number of poor people was 25.47 thousand people with a percentage of 13.47%. Looking at the data, there was a decrease in the poverty rate of 0.34%. This was in line with the Head of Institutional and Community Training (DPMD) interview results below:

*"Yes, it can alleviate poverty; at least it helped the community's economy"*.

From the analysis above, for indicators of outcomes, the village fund policy has had many impacts on village communities, especially community welfare and development. Public services in the village increased but were not optimal; this was because not all village offices were active every day, the facilities and infrastructures in the village office were inadequate, and the competence of public service providers often changed policies and regulations from the government, reducing poverty, increasing people's income and absorbing more labor.

This research was designed for sustainable community development in reducing the percentage of poverty at the village level. Most of the poor live in rural areas, so Indonesia must fight poverty from its roots first [40,41]. This was reviewed from the four indicators related to previous research, which explained that village funds are always associated with the economy, society and ecology [5]. Previous research explained that village funds were also used to increase knowledge about sanitation because there are still many rural areas in Indonesia that do not pay attention to health regulations or programs [42]. This might be

used as a consideration in conducting sustainable research on village funds on sanitation regulations, meaning that healthy villages will produce healthy and intelligent residents. Village funds in the PALI district are still focused on being used for road facilities and infrastructure and therefore do not have much impact in reducing poverty. This statement is based on previous research by Hermawan et al. (2019) [43]. This village still needs to catch up to other villages that have focused on allocating village funds for educational purposes. The quality of education in an area impacts the nation's future because it is the primary key that drives national change. The PALI district government should be able to learn from other districts which have now improved the quality of education and number of literate people [44].

These four indicators (inputs, processes, outputs and outcomes) provide knowledge about the state of village funds in several villages in the PALI district. This is an illustration for the government of what needs to be considered to improve village conditions towards developed villages so that village funds can continue beyond infrastructure development.

## 6. Conclusions

The evaluation of village fund policies in Penukal Abab Lematang Ilir (PALI) concluded that the evaluation of village fund policies in Penukal Abab Lematang Ilir (PALI) in terms of input, process, output and outcome indicators had not gone well. Input indicators such as regulations were complete but seemed to overlap; aspects of the adequacy of funds and village funds met the needs of rural communities, while the aspects of facilities, infrastructure and human resources were insufficient. Process indicators such as planning, implementation, and monitoring of the implementation using village funds were appropriate. However, there were several obstacles in making reports using the Siskeudes application, namely limited skills in using technological tools, so operators were needed. Output indicators were implemented according to existing regulations but only focused on physical development. Outcomes from the evaluation of village funds have had many impacts, especially in reducing the percentage of poverty. Village heads must control regulations, and village officials and local governments must be expected to monitor the transparency of village funds more closely and focus on other challenges, not only in infrastructure but also in educational development. Based on the research conducted, there were several weaknesses found in this study, one of which was that the researcher conducted interviews with a limited number of informants so that the researcher could not comprehensively analyze the evaluation of the village fund policy that had been carried out in the Penukal Abab Lematang Ilir regency. Therefore, future research is expected to expand the involvement of other informants, and then triangulation can be carried out more clearly and measurably and can use other evaluation theories to determine the village fund policy evaluation results. The other weakness is that it is necessary to conduct further studies on the methods used to evaluate the village fund policy. This is intended to re-examine the results of this study to make improvements to the results of the study on the evaluation of the village fund policy, and to develop concepts related to the evaluation of the village fund policy. It can be concluded that the village fund helps improve the village economy and will make a better contribution if a village and everyone who works in it program a smart village. A smart village means that, in managing information technology and conditions in technology development, the residents are increasingly aware of and willing to change following developments in managing village funds, but must be programmed.

*Practical Suggestions*

1. The village Community Empowerment Service needs to continue to evaluate the village fund policy in the Penukal Abab Lematang Ilir regency. It is intended that this village fund policy can continue to be improved and provide solutions to evaluation problems.
2. In implementing the village fund policy, several regulations must be controlled by the village head and village apparatus. However, the regulations overlap, and the policy regulations often change. In dealing with regulations, village officials must also

be good at using technology because village fund management uses the Siskeudes application. To overcome this, the regional and central governments need to improve the human resources of the village apparatus, either through formal education or through training and technical guidance. In addition to going through the guidance process, the central government needs to redesign technical regulations related to village heads' and officials' requirements.

3.  The level of community participation in the implementation of the village fund policy is still relatively low, so the local government must intensify socialization with the community regarding the village fund policy and priorities for the use of the village fund so that the community does not misunderstand the use of village funds.

4.  Local governments are expected to supervise more closely for transparency of village funds, such as the installation of billboards/banners.

**Author Contributions:** Conceptualization, S.; methodology, S.; formal analysis, S.; investigation, S.; writing—original draft preparation, S.; writing—review and editing, S.; project administration, S., E.AM., R.A.B. and D.; supervision, S., E.AM., R.A.B. and D. All authors have read and agreed to the published version of the manuscript.

**Funding:** This research was funded by the Education Fund Management Institution of Lembaga Pengelola Dana Pendidikan (LPDP), (No: 20200421301016 for Suandi). The APC was funded by Universitas Padjadjaran.

**Institutional Review Board Statement:** Not applicable.

**Informed Consent Statement:** Informed consent was obtained from all subjects involved in the study.

**Data Availability Statement:** Not applicable.

**Acknowledgments:** The authors are thankful for the contribution of all informants interviewed for this study. Finally, the authors also thank the anonymous reviewers for their valuable reviews and suggestions. The authors thank the Directorat Riset Pengabdian Masyarakat (DRPM) and Universitas Padjadjaran for funding the APC and Lembaga Pengelola Dana Pendidikan (LPDP) for supporting this study.

**Conflicts of Interest:** The authors declare no conflict of interest.

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
