# Peer review of "The Evaluation of Village Fund Policy in Penukal Abab Lematang Ilir Regency (PALI), South Sumatera, Indonesia"

_sustainability, doi:10.3390/su142215244_

Round 1

Reviewer 1 Report

The topic of your study is clearly of interest. Thanks for all your work done.

Editing aspects:

there was abuse[6] => there was abuse [6]

If it is the short form for Local Development Funds, please type it in brackets when using it the first time: Local Development Funds (LDFs)

Based on the preliminary research that the researcher has done
"the researcher" refers to the authors of this publication or someone else?

Indicator Outputs, namely Output Indicators, => would not just Outbput Indicators ok?

Recommendations:

Adding some sentences about the limitations (especially in case of the method applied)/merit if this contribution

It would increase the value of the contribution if the concept of SMART VILLAGES is considered as well.

References: line 38 and 39: ";" after the year

Author Response

Point 1: There was abuse[6] => there was abuse [6]

Response 1: We already revised to be This policy has been implemented worldwide, including in Thailand, which provided microcredit to the poorer, agricultural and households but there was abuse in its implementation [6].

Point 2: If it is the short form for Local Development Funds, please type it in brackets when using it the first time: Local Development Funds (LDFs)

Response 2: We already revised

Point 3: Based on the preliminary research that the researcher has done
"the researcher" refers to the authors of this publication or someone else?

Response 3: The researcher refers to the author and we already explained that

Point 4: Indicator Outputs, namely Output Indicators, => would not just Outbput Indicators ok?
"the researcher" refers to the authors of this publication or someone else?

Response 4: We already deleted one, and we changed with Output Indicators

Point 5: Adding some sentences about the limitations (especially in case of the method applied)/merit if this contribution

Response 5: We already added in the conclusion

Point 6: It would increase the value of the contribution if the concept of SMART VILLAGES is considered as well.

Response 6: We already added some sentence of the smart villages context

Point 7: References: line 38 and 39: ";" after the year

Response 7: the references refers to [3,4]

Reviewer 2 Report

The authors state (lines 113 – 116): “It was necessary to evaluate the village fund policy to improve rural communities' welfare. So, this research was more about reviewing and evaluating village fund policies, including input, process, output, and outcomes.”

The manuscript looks to be interesting, but it substantially suffers. Its major weaknesses are listed below.

- The objectives of the research are not clear and it is difficult for the reader to understand what exactly is being examined.

- The methodology the authors use is not clearly presented and it takes a lot of effort to understand exactly what the research is looking at and how. For instance, in lines 147-148 the authors note that “... the evaluation measurement used criteria consisting of input, process, output and outcomes.”, but sufficient clarifications aren't given. The same stands for the lines 139-146. The authors should present the methodology in a way that enables the replicability of the research.

- The number of interviewees is too small and it can raise doubts about the accuracy of the research results and the generalizability of the conclusions.

- A critical review of the literature is not presented to better illustrate the contribution of the present research to the field.

- There is no a deep discussion of the results of the research, in which the results of the present research are compared with the findings of other researchers in the same country and worldwide.

- The implications the authors note are too general. They should make them more specific such as the implications for public authorities at national, regional and local level as well as for other stakeholders and academics…

- In lines 266-267, the authors note: “Based on the data analysis above, the adequacy of funds, it can be judged that village funds were sufficient for village activities.” It isn't clearly defined which data and how these are analysed.

- In the results section, the authors write their findings but they often use the phrase “must be done...". These phrases complicate the situation and thus, additional ambiguities arise about the interviewer's questions and the interviewees' answers.

- The authors present some arithmetic data without specifying what these data express/reflect.

- They present some abbreviations without define them in the first time, as the journal guidelines require.

- Some elements, including also photos, that do not add anything to the article are included in the text. These can be removed or included in an appendix.

- There is no a deep discussion of the results of the study in comparison to those of previously published studies (as well as methodologies).

- They authors should also improve some phrases they use, simplify the language style they use as well as add some more bibliographic sources related to the present research.

- There are also some unnecessary repetitions, and many names are used which confuse the reader.

The authors should elaborate the manuscript following the above reported comments. They could also make it more friendly to the reader.

Author Response

Thank you for the comment and suggestion, we already revised and improved it.

Reviewer 3 Report

Dear authors,

The role of public funds in the development of rural zones is essential. In the paper you have submitted you have shown the variety of challenges that Penukal Abab Lematang Ilir Regency needs to face in order to increase its socioeconomic development. However, this paper needs major changes in order to be considered for publication in this journal:

- The introduction section introduces basic features of the region, but it is in the third section where more information about it is provided. This is very confusing, and you do not even mention the main economic activities of the region. 

- There is a paragraph in the introduction section that it is repeated in the results section (lines 85 to 92). Delete one of them.

- The Definition section is very poorly made. I suggest changing it into a "Literature review"section in which you show what has been researched about the relevance of public policies in the development of rural areas. 

- The results section is confusing as well. I do not understand whether the points following 5.1. Input indicators are indeed indicators. If so this should be better structured in the paper. 

- During this review I have been referring to the results section instead of the name you have used in the paper (results and discussion). I see you have not included any discussion of the results. You should compare these results with similar experiences, trying to learn from them and apply these learnings to your region. 

- Conclusions are just a summary of the results. You should include future lines of investigation as well as the limitations of the current research.

Author Response

Response to Reviewer 3 Comments

Point 1: The introduction section introduces basic features of the region, but it is in the third section where more information about it is provided. This is very confusing, and you do not even mention the main economic activities of the region.

Response 1: We already checked and added the information in the third section.

The area of ​​​​Penukal Abab Lematang Ilir (PALI) is classified as agrarian, with average rainfall in 2019 varying between 25 mm to 320 mm, where the highest rainfall occurs in February. There are four types of soil in Penukal Abab Lematang Ilir (PALI): alluvial, red-yellow podzolic, gley association, and yellowish-brown podzolic association. These four types of soil are found in almost all sub-districts in Penukal Abab Lematang Ilir (PALI) Regency, except for yellowish brown podlosik association soil, which is only found in Penukal District.

Based on the results of the population census conducted annually, the population of the Penukal Abab Lematang Ilir (PALI) Regency was recorded at 187,281 people in 2019. It is 189,764 people, and in 2020 it is 192,199 people (Central Bureau of Statistics of PALI Regency, 2021). From this data, it can be seen that in 2019 the population growth rate in the Penukal Abab Lematang Ilir (PALI) Regency was recorded at 1.32 percent compared to the previous year. This indicates an increase in the population in this region, from 187,281 people in 2018 to 189,764 people in 2019. In addition, population growth in 2019 experienced a slowdown compared to the previous year, with a growth rate of 1.32 percent. Most of the population (41.92%) live in the district capital, Talang Ubi.

Furthermore, when viewed from the age group, the population of children (0-14 Years) in the Penukal Abab Lematang Ilir (PALI) Regency in 2018 reached 303.05%, while the productive age population reached 62.8%. The population aged further 4.12% of the total population in the Regency of Penukal Abab Lematang Ilir (PALI). Likewise, in 2019, the child population had a percentage of 32.8%, productive age reached 62.9% and older adults at 4.2%. It can be seen that the population of the Penukal Abab Lematang Ilir (PALI) Regency is generally productive. The composition of a population like this is beneficial if the population is qualified and can contribute to development. Vice versa can also be detrimental if the population is of low quality. Therefore the government of Penukal Abab Lematang Ilir (PALI) must provide maximum education because only then can the population be used to the maximum. Besides that, it can also be stated that the overall male population in Penukal Abab Lematang Ilir (PALI) Regency is more than the female population.

The economic structure in Penukal Abab Lematang Ilir (PALI) in 2019 was still dominated by the primary category, particularly the Mining and Quarrying category (43.20 percent) and the Agriculture, Forestry and Fisheries category (15.24 percent). While the second largest role comes from the tertiary category, which comes from the wholesale and retail trade category; repair of cars and motorcycles (14,65). While in the tertiary category, the most considerable role comes from the construction category (14.34) (Central Bureau of Statistics of PALI Regency, 2020).

The PMK Number 49/PMK.07/2016, evaluation of the Village Fund policy was carried out on the calculation of the distribution of the amount of the Village Fund for each Village by the Regency/City and the realization of the distribution and use of the Village Fund and assessed based on appropriate inputs, processes, outputs, and outcomes and evaluated based on proper inputs, processes, outputs, and outcomes [31]

Point 2: There is a paragraph in the introduction section that it is repeated in the results section (lines 85 to 92). Delete one of them

Response 2: We already deleted one of them.

Point 3: The Definition section is very poorly made. I suggest changing it into a "Literature review"section in which you show what has been researched about the relevance of public policies in the development of rural areas

Response 3: We already changed to literature review.

Point 4: The results section is confusing as well. I do not understand whether the points following 5.1. Input indicators are indeed indicators. If so this should be better structured in the paper.

Response 4: We have checked and revised the information for a better and more structured presentation before explaining each indicator below.

“Then in 2019, this village fund increased to 77.19 billion following the Central Government's policy to increase village funds.

  1. District Inspectorate;
  2. Community and Village Empowerment Service (DPMD);
  3. District.

The village fund policy evaluation is carried out based on a portfolio assessment by desk evaluation and field assessment. The portfolio assessment is carried out based on the data contained in the Village Fund Policy Consolidation Report. Meanwhile, the field assessment by the evaluator team was carried out by directly observing the villages that received the Village Fund disbursement.

Specifically for the evaluation of the Village Fund policy in the Penukal Abab Lematang Ilir Regency, the results of the village fund evaluation show that the Penukal Abab Lematang Ilir Regency has not been optimal in implementing the village fund policy. The evaluation results require clarification and more objective evaluation efforts, considering that several weaknesses and problems are still being found, especially when viewed from the aspect of substance, implementation and results of village fund policies in improving community welfare.

In this section, the researcher will describe the results of the field research and its discussion. In this discussion, the theory used as an analytical tool or guidance is the theory of Bridgman and Davis (Bridgman et al., 2020) with several indicators, namely input, process, output, and outcomes. Furthermore, these indicators serve as a guide for researchers in digging up field data and discussing research results.”

Point 5: During this review I have been referring to the results section instead of the name you have used in the paper (results and discussion). I see you have not included any discussion of the results. You should compare these results with similar experiences, trying to learn from them and apply these learnings to your region.

Response 5: Thanks for the suggestion, we already elaborated the discussion with the several previous researchs. “This research was designed for sustainable community development in reducing the percentage of poverty at the village level. Most of the poor live in rural areas, so Indonesia must fight poverty from its roots first [41,42]. This was reviewed from the four indicators related to previous research, which explained that village funds are always related to the economy, society and ecology [5]. Previous research explained that village funds were also used as assistance to increase knowledge about sanitation because we know that there are still many rural areas in Indonesia that do not pay attention to health regulations or programs [43]. This might be used as a consideration in conducting sustainable research on village funds in the future on sanitation regulations, and healthy villages will produce healthy and intelligent residents. Village funds in the PALI district are still focused on being used for road facilities and infrastructure so that village funds do not have much impact in reducing poverty. This statement is under previous research conducted by Hermawan et al., 2019[44]. This village still needs to catch up to other villages that have focused on allocating village funds for educational purposes. The quality of education in an area impacts the nation's future because it is the primary key that drives national change. The PALI district government should be able to learn from other districts which have now improved the quality of education and literate people [45].

These four indicators (inputs, processes, outputs and outcomes) provide knowledge about the state of village funds in several villages in the PALI district. This is an illustration for the government of what needs to be considered to improve village conditions towards developed villages so that village funds continue beyond infrastructure development.

Point 6: Conclusions are just a summary of the results. You should include future lines of investigation as well as the limitations of the current research

Response 6: we already included the future lines of investigation as well as the limitations of the current research. “The evaluation of village fund policies in Penukal Abab Lematang Ilir (PALI) is concluded that the evaluation of village fund policies in Penukal Abab Lematang Ilir (PALI) in terms of input, process, output and outcome indicators had not gone well. Input indicators such as regulations were complete but seem to overlap; aspects of the adequacy of funds and village funds have met the needs of rural communities, while the aspects of facilities, infrastructure and human resources were insufficient. Process indicators such as planning, implementation, and monitoring of the implementation using village funds were appropriate. However, there were several obstacles in making reports using the Siskeudes application, namely limited skills in using technological tools, so operators were needed. Output indicators have been implemented according to existing regulations but only focus on physical development. Outcomes from the evaluation of village funds have had many impacts, especially in reducing the percentage of poverty. Village heads must control regulations, and village officials and local governments were expected to monitor the transparency of village funds more closely and needed to focus in other challenges not only in infrastructure but also in educational development. Based on the research that had been done, there were several weaknesses found in this study, one of which was that the researcher conducted interviews with a limited number of informants so that the researcher could not comprehensively analyze the evaluation of the Village Fund policy that had been carried out in Penukal Abab Lematang Ilir Regency. So, the future research is expected to expand the involvement of other informants then triangulation can be carried out more clearly and measurably and can use other evaluation theories to determine the village fund policy evaluation results. The other weakness is that it is necessary to conduct further studies on the methods used to evaluate the Village fund policy further. This is intended to re-examine the results of this study to make improvements to the results of the study on the evaluation of the Village Fund policy and to develop concepts related to the evaluation of the Village Fund policy. It can be concluded that the Village Fund helps improve the village economy and will make a better contribution if a village and everyone who works program a smart village. Smart village means that in managing information, technology and conditions in technology development, they are increasingly aware of and willing to change following developments in managing village funds but must be programmed.

6.1. Practical Suggestions

  1. The Village Community Empowerment Service needs to continue to evaluate the Village fund policy in Penukal Abab Lematang Ilir Regency. It is intended that this village fund policy can continue to be improved and provide solutions to evaluation problems.
  2. In implementing the village fund policy, several regulations/regulations must be controlled by the village head and village apparatus. However, the regulations overlap, and the policy regulations often change. In dealing with regulations, village officials must also be good at using technology because village fund management uses the Siskeudes Application. To overcome this, the Regional and Central Governments need to improve the human resources of the village apparatus, either through formal education or through training and technical guidance. In addition to going through the guidance process, the central government needs to redesign technical regulations related to village heads and officials' requirements.
  3. The level of community participation in the implementation of the village fund policy is still relatively low, so the local government must intensify socialization with the community regarding the village fund policy and priorities for the use of the village fund so that the community does not misunderstand the use of village funds.
  4. Local governments are expected to supervise more closely for transparency of village funds, such as the installation of billboards/banners.”

Round 2

Reviewer 2 Report

Dear Editors

The authors elaborated on the manuscript and made improvements, following my most important comments and suggestions. They have also responded to my comments and suggestions, point by point, as the journal guidelines require.

The only comment I have to make is that the implications the authors write do not need to be numbered.

Thus, Ι suggest the manuscript to be accepted for publication.

Reviewer 3 Report

Dear authors,

I have checked the revised version of the manuscript and I have no further objections regarding the publication of your paper in this journal.